# Pharmacogenomic Analysis of Combined Therapies against Glioblastoma Based on Cell Markers from Single-Cell Sequencing

**DOI:** 10.3390/ph16111533

**Published:** 2023-10-30

**Authors:** Junying Liu, Ruixin Wu, Shouli Yuan, Robbie Kelleher, Siying Chen, Rongfeng Chen, Tao Zhang, Ismael Obaidi, Helen Sheridan

**Affiliations:** 1NatPro Center, School of Pharmacy and Pharmaceutical Sciences, Trinity College Dublin, D02 PN40 Dublin, Ireland; tao.zhang@tudublin.ie (T.Z.); obaidii@tcd.ie (I.O.); hsheridn@tcd.ie (H.S.); 2Preclinical Department, Shanghai Municipal Hospital of Traditional Chinese Medicine, Shanghai University of Traditional Chinese Medicine, No. 274, Zhijiang Road, Jing’an District, Shanghai 200071, China; rxwu2000@hotmail.com; 3Academy for Advanced Interdisciplinary Studies, Peking University, Beijing 100871, China; yuanshouli123@163.com; 4School of Medicine, Trinity College Dublin, D02 PN40 Dublin, Ireland; kellehro@tcd.ie; 5The Second Affiliated Hospital, Nanchang University, Nanchang 330031, China; shchen886@outlook.com; 6National Center for Occupational Safety and Health, NHC, Beijing 102308, China; rongfeng.chen2009@outlook.com; 7School of Food Science & Environmental Health, Technological University Dublin, D07 EWV4 Dublin, Ireland

**Keywords:** glioblastoma, pyroptosis, scRNA-seq, systems pharmacology, CADD, GBM, pharmacogenomics

## Abstract

Glioblastoma is the most common and aggressive form of primary brain cancer and the lack of viable treatment options has created an urgency to develop novel treatments. Personalized or predictive medicine is still in its infancy stage at present. This research aimed to discover biomarkers to inform disease progression and to develop personalized prophylactic and therapeutic strategies by combining state-of-the-art technologies such as single-cell RNA sequencing, systems pharmacology, and a polypharmacological approach. As predicted in the pyroptosis-related gene (PRG) transcription factor (TF) microRNA (miRNA) regulatory network, TP53 was the hub gene in the pyroptosis process in glioblastoma (GBM). A LASSO Cox regression model of pyroptosis-related genes was built to accurately and conveniently predict the one-, two-, and three-year overall survival rates of GBM patients. The top-scoring five natural compounds were parthenolide, rutin, baeomycesic acid, luteolin, and kaempferol, which have NFKB inhibition, antioxidant, lipoxygenase inhibition, glucosidase inhibition, and estrogen receptor agonism properties, respectively. In contrast, the analysis of the cell-type-specific differential expression-related targets of natural compounds showed that the top five subtype cells targeted by natural compounds were endothelial cells, microglia/macrophages, oligodendrocytes, dendritic cells, and neutrophil cells. The current approach—using the pharmacogenomic analysis of combined therapies—serves as a model for novel personalized therapeutic strategies for GBM treatment.

## 1. Introduction

Glioblastoma (GBM) is the most common and aggressive form of brain cancer, with no known cure or prevention [1]. GBM has the worst prognosis of any brain cancer, with an overall survival rate of approximately three months if untreated and around 14–20 months if treated with surgery, chemotherapy, and radiotherapy. GBM is prone to metastasis and high rates of post-treatment progression [2]. The primary reason for its poor prognosis lies in the diffusion of highly invasive individual tumor cells through the brain parenchyma, coupled with the resilience of the brain tumor progenitor cells [3]. A huge unmet clinical need is the full understanding of the pathogenesis and related pathways of GBM.

Advances in various research areas have brought new insights and therapeutic perspectives to GBM research. Single-cell RNA sequencing (scRNA-seq) offers advantages for cell-type characterization and cell-type heterogeneities based on the dynamic gene expression of each cell and it is widely applied in cancer and immunology research [4]. The Human Cell Atlas (THCA) represents one of the recent rapid technological advances in the single-cell analysis community; it aims to describe each human cell by the expression level of approximately 20,000 human protein-coding genes. Therefore, compared with bulk sequencing, scRNA-seq unveils the complex gene expression at the single-cell level and cell heterogeneity in complex tissues, providing specific information on tumor microenvironments and drug resistance [5]. However, a bulk analysis of the whole tumor may be a hurdle to understand, particularly the complicated GBM ecosystem-driven infiltration. In contrast, a single-cell analysis may provide detailed information on relevant events. Moreover, single-cell level analyses of the microenvironment may depict tumor invasion due to the immunosuppressive event blocking effective T cell immune responses [6]. 

The aggressive and incurable nature of GBM has created an urgency for the development of novel therapeutic approaches. The only approved medicinal drug, the benzimidazole, only succeeds in slowing down, but not terminating, tumor growth and often fails due to side effects and drug resistance [7]. Drug repurposing could be an efficient strategy to test approved or investigational drugs for new uses with the distinct advantage of a lower overall cost and less time than developing an entirely new drug from scratch [7,8]. Developing specific inhibitors to counteract the detrimental impacts of mutated oncogenic proteins can be challenging because most current anticancer drug targets are in both normal and cancerous cells [9]. Nevertheless, cumulative evidence has lent confidence to discovering drug repurposing candidates *via* the Connectivity Map (CMap) platform [10]. As an appealing complementary tool to phenotype-based drug screening for lead molecules, CMap is a useful exploratory tool to identify novel bioactive compounds, with therapeutically beneficial results from a diseased tissue sample with a transcriptomics dataset. These novel compounds can be used for pharmaceutical pipelines across disease disciplines [10]. Major hurdles in current GBM treatment include poor clinical outcomes, flawed treatment strategies, and unwanted adverse side effects. In contrast, naturally derived lead molecules, with positive anticancer effects and minimal side effects, may present a better treatment option to be considered [11]. 

More than 50% of chemically synthesized drugs are derivatives from plant isolates [12]. The predictive toxicology of these plant metabolites has made great progress due to advances in computational approaches, systems biology, and pharmacogenomic analyses [2]. As an innovative strategy in drug discovery, polypharmacology can expedite drug exploration by integrating systems biology and pharmacology by deploying biology, chemistry, and systems modeling across human diseases [13]. Two aspects need to be considered for a polypharmacological approach with a potentially beneficial therapeutic result: (1) an effective drug candidate that targets multiple proteins; and (2) several drug candidates targeted at one hub protein, leading to the activation of multiple signaling and functional pathways [12]. However, a polypharmacological strategy faces two challenges: (1) the identification of a target protein/combined targets in a disease sample with a predictable response to drug perturbation; and (2) the discovery of a multi-target molecule with the desired polypharmacological profile to perturb those targets [12]. In this sense, systems pharmacology likely estimates the matching modes between the combined drugs and cell complexity in pathological states to decode the drug’s mechanisms of action (MOA) by integrating systems biology, pharmacokinetics, and pharmacodynamic methods. The definition of clinical and genetic alterations within methylation classes has allowed the identification of numerous new diagnostic models for a personalized approach to patient care [14]. Not all individuals are in an identical disease state. Therefore, future therapeutic strategies should move toward more personalized and targeted therapies based on highly sensitive and accurate biomarkers. This could lead to a successful application of an omics-based personalized clinical approach. Although the genome-wide investigation of driver mutation has provided a clear road map to guide precision and customized cancer drugs [15], intratumoral heterogeneity remains a considerable challenge. Fortunately, single-cell technologies have been developed to meet this challenge [16]. Although drug development is a fast-developing area, multi-target drug discovery as personalized and predictive medicine is still in its infancy stage at present. This research aimed to discover biomarkers to inform disease progression and develop personalized prophylactic and therapeutic strategies by combining state-of-the-art technologies such as scRNA-seq, The Cancer Genome Atlas (TCGA) cohort, systems pharmacology, and computer-aided drug design (CADD), which could serve as a new model for the conception and design of novel personalized therapeutic approaches for cancer treatment. 

## 2. Results

### 2.1. Single-Cell Sequencing Analysis

The R package Seurat could separate the GBM cells into 15 clusters based on differentially expressed genes (DEGs) after a QC analysis and PCA (Appendix A). The data were visualized using a t-SNE plot in a two-dimensional projection space. A set of signature marker genes for each cluster was identified to define the cell types. The cluster numbers were assigned from the largest cell (Cluster 0) to the smallest (Cluster 14). By identifying highly enriched marker genes for each cluster, the cells were classified into 15 cell types as well as the number of cells and characteristic genes in each cluster. The Cluster 0, Cluster 2, and Cluster 7 cells were microglia-expressing signature genes, including P2RY12, P2RY13, SLC1A3, and CX3CR1. Cluster 1 and 13 were different subtypes of microglia or macrophages expressing SLC1A3 and DAB2, respectively. Cluster 3 characterized macrophage cells with high CD163 and APOC1, whereas Cluster 9 represented macrophage cells expressing CD163, F13A1, and APOC1. Cluster 4 represented B cells expressing CD52. Cluster 5 represented glia and neuronal cells that expressed CCL4. Cluster 6 represented proliferating macrophages expressing PKM and PFN1. Cluster 8 comprised mural cells expressing CALD1 and IGFBP7. Cluster 10 comprised neutrophils expressing FPR2, IL1R2, and CSF3R. Cluster 11 represented dendritic cells expressing CD1C, FCER1A, CD1E, HLA-DPB1, and HLA-DQA1. Cluster 12 comprised T cells with signature genes of CD48. Cluster 14 represented endothelial cells expressing CAVIN2, VWF, ABCG2, and CLDN5. 

### 2.2. Pyroptosis-Related Gene Analysis

The 185 pyroptosis-related genes were collected from GeneCards with screening criteria of a relevance score > 1 (102 genes) (Appendix A). The intersection yielded 30 PRGs in GBM samples for mutation frequency, oncoplot waterfall plot, and PRG-TF-miRNA regulatory network (Figure 1A). The PRG-TF-miRNA regulatory network showed that TP53 was the hub gene in the related PRG network. The gene expression profile across all tumor samples and paired normal tissues is presented in a dot plot in Figure 1B. The mutation distribution and protein domains of TP53 in the GBM samples are labeled as current hotspots on the lollipop plot in Figure 1C. The plot title and subtitle represent the somatic mutation rate and transcript name. The oncoplot in Figure 1D demonstrates the somatic landscape of the TP53 cohort, where the x-axis shows the mutation frequency of the genes and the disease tissues are ordered by an annotation bar (bottom). The oncoplot shows the gene mutation information from each sample; the different colors with specific annotations at the bottom represent different mutation types. A cohort summary plot represents the distribution of variants according to the variant classification, type, and single nucleotide variant (SNV) class. A stacked bar plot shows ten mutated genes. The bottom part, from left to right, shows the mutation load for each sample and variant classification type (Figure 1E). The overall expression of the 30 PRGs from three different brain tissues (normal, GBM, and low-grade gliomas (LGGs)) from the TCGA cohort was analyzed. APOE, TP53, CASP6, CASP8, and DHX9 showed a high expression in the tumor tissues (Figure 1F). The clustering of 30 genes in the PPI using the K-means clustering algorithm showed that CASP1, IL18, and IL1B were the hub genes of one cluster and TP53 was the hub gene of another cluster (Appendix A).

### 2.3. The Prognostic Value of Pyroptosis-Related Genes

The prognostic value of pyroptosis-related genes with the top eight HR values in GBM patients in the high-/low-expression groups was 5.18 for CASP6, 5.17 for GBP1, 4.73 for CASP8, 4.24 for CASP4, 4.1 for GBP5, and 4.07 for GSDMD (Figure 2). The coefficients of selected features were shown using the lambda parameter and a partial likelihood deviance versus log (λ) was drawn using the LASSO Cox regression model (Equation (1)). The model was a risk factor if HR > 1 and the model was a protective factor if HR < 1; a 95% CI represented the HR confidence interval. The median time or the median survival time represented the survival rate of the two groups, corresponding with 50% of the time in units of years. For example, the survival times in the high- and low-expression groups were 1.7 years and 8 years, respectively (Figure 3). The higher the area under the curve (AUC) value, the stronger the model’s predictive ability. The dotted line represented the median risk score and divided the patients into low-risk and high-risk groups. A greater number of dead patients corresponded with a higher risk score. A heatmap of the expression profiles of the prognostic genes in the low- and high-risk groups was produced accordingly. The time-dependent receiver operator characteristic (ROC) curve and AUC showed that the higher the AUC value, the stronger the predictive ability of these genes. The AUCs were 0.857, 0.892, and 0.815 for 1 year, 2 years, and 3 years, respectively; all were greater than the predictive threshold of 0.5 and the accurate prediction threshold of 0.7. 

The LASSO model was as follows: Lambda.min = 0.0047  Riskscore = (0.4066) × CASP6 + (0.0316) × GBP1 + (0.7417) × CASP4 + (0.1422) × GBP5 + (0.1563) × GSDMD + (0.2342) × CASP3 + (−0.1911) × CASP1 (1)

The HR and *p*-value of the constituents were involved in univariate and multivariate Cox regressions as well as some parameters of the top five PRGs. The nomogram to predict the 1-, 2-, and 3-year overall survival of GBM patients was based on the risk signature CASP4 and age (Figure 4). The line segment corresponding with each variable in the nomogram was marked with a scale that represented the value range of the variable; the length of the line segment reflected the factor’s contribution to the outcome event. The nomogram’s prediction efficiency demonstrated that the model’s C-index was 0.619 (CI: 0.544–1) with a *p*-value of 0.002, suggesting the moderate accuracy of this predictive model. In addition, a calibration plot showed that the bias-corrected line was close to the ideal curve (45 degree line), suggesting a satisfactory agreement between the prediction and the observation. Therefore, this nomogram could be a clinically useful tool for prognostic predictions in GBM patients.

### 2.4. The Abundance of Immune Cells and TMB in GBM

The correlation of tumor purity as well as the expression of the top eight HR genes with six infiltrating immune cells (including CD8^+^ T cells, CD4^+^ T cells, B cells, dendritic cells, macrophages, and neutrophils) (Appendix A) were analyzed. Based on the TIMER algorithm, the infiltration level of immune cells was positively and significantly correlated with the expression level of these genes using the Spearman correlation at *p* < 0.05 (Appendix A). The x-axis represented the gene expression distribution and the y-axis represented expression distribution. The correlation coefficient ranged from −1 to 1; negative numbers indicated a negative correlation between two genes. The closer the value was to 1 or −1, the stronger the correlation between the immune cells and PRG expression; the closer to 0, the weaker the correlation between them. These findings suggested that these PRGs could trigger a better immune response than other genes. 

### 2.5. Drug Prediction, Drug Sensitivity, and Drug Validation

The cancer therapeutic drug prediction from the relationship between diseases and the drug database (DSigDB) as well as the half inhibitory concentration (IC50) analyses of these repurposed drugs are shown in Figure 5A. A Sankey plot showcasing the drugs repurposed to the pyroptosis-related phenotype of GBM showed that 20 out of 30 genes were involved in each repurposed drug obtained via the DSigDB database. The dot plot showed the ratio between pyroptosis-related genes specific to repurposed drugs and the total number of genes targeted by each drug (false-discovery rates (FDRs); *p* < 0.05). IC50 is an important indicator for the evaluation of drug efficacy or sample treatment responses. This tool is based on the largest public pharmacogenomic database, Cancer Drug Sensitivity Genomics (GDSC). The GDSC drug sensitivity analysis of GBM showed that the IC50 values for vorinostat, 5-fluorouracil, pyrimethamine, gefitinib, tamoxifen, and sorafenib tosylate were 8.25 mM, 2.80 mM, 3.20 mM, 1.19 mM, 3.3 mM, and 1.75 mM, respectively. The natural compounds were selected from the drug prediction results of CMap (Figure 5B). The potential compounds with therapeutic benefits were obtained by querying the CMap database with the GBM subgroup samples’ signature expression. The top 21 natural molecules with negative connectivity scores were selected from 49,313 agents to repress the signature gene expression profile of each cell cluster of GBM (Figure 6). The top 5 of the 21 natural compounds were parthenolide, rutin, baeomycesic acid, luteolin, and kaempferol, with NFKB inhibitor, antioxidant, lipoxygenase inhibitor, glucosidase inhibitor, and estrogen receptor agonist MOAs, respectively (Table 1). In contrast, the analysis of the cell-type-specific differential expression-related targets of the 21 natural compounds showed that the top 5 subtype cells targeted by natural compounds were endothelial cells, microglia/macrophages, T cells, dendritic cells, and neutrophil cells.

## 3. Discussion

As a newly discovered programmed cell death (PCD) system, pyroptosis is a hot topic in cancer initiation and progression [17]. This type of PCD is also called secondary necrosis due to the complete apoptosis process of releasing inflammatory mediators. The macrophages can phagocytize the initiated apoptotic cells and if macrophages cannot phagocytize the apoptotic cells, secondary necrosis occurs. Consequently, a series of inflammatory responses arise [17]. The complex relationship between pyroptosis and cancer has attracted increased attention because pyroptosis can provide a favorable microenvironment for tumor proliferation. In contrast, excessive pyroptosis activation inhibits tumor cells [18]. Therefore, the dual effects of pyroptosis on tumors may help to develop an important cancer therapy strategy. Recent studies on the relationship between pyroptosis and gastric cancer, breast cancer, and GBM have provided new research ideas for cancer prevention and treatment [19]. In this study, as predicted in the PRG-TF-miRNA regulatory network, TP53 was the hub gene in pyroptosis-related GBM, which is common across tumor types [20]. Common tumor cell variants generally include SNVs, multiple nucleotide variants (MNVs), insertion, deletion, complex variants, and structural variants (SVs). The popular software used for variant calling, such as Genome Analysis Toolkit (GATK, Broad Institute, gatk-4.4.0.0), FreeBayes (https://github.com/freebayes/freebayes (accessed on 18 October 2023)), and VarScan (https://varscan.sourceforge.net/using-varscan.html (accessed on 18 October 2023)), was designed to detect SNVs, small insertions, and deletions, but not for complex variants. However, tumor-suppressed genes such as TP53, PTEN, BRA1/2, RB1, STK11, and NF1 often consist of large fragments of frameshift insertions, deletions, or complex variants as well as SVs, which are often missed by detection software [21]. In this study, the SNV analysis showed that TP53 was one of the most frequently mutated genes in GBM. Therefore, it could serve as a biomarker of particular molecular characteristics and a prognostic tool for unfavorable survival in GBM [22]. 

It is essential to tailor the specialized management of GBM patients. Therefore, we also investigated the prognostic value of other PRGs by constructing a LASSO Cox model. The LASSO Cox regression model of six pyroptosis-related genes (CASP6, CASP8, CASP4, GBP1, GBP5, and GSDMD) was built to accurately and conveniently predict the 1-, 2-, and 3-year overall survival rates of GBM patients based on the dataset from TCGA. Previous research demonstrated that GBP5 is a driving factor for GBM malignancy *via* the Src/ERK1/2/MMP3 pathway and its high expression may represent a poor prognosis in GBM [23]. Our result showed that GBP5’s HR was high, consistent with previous research.

Moreover, the multivariate Cox model showed that CASP4 had the highest HR ratio, indicating that CASP4 could be an independent prognostic parameter for GBM patients. Following this, a further exploration of the role of CASP4 in the prognosis of GBM patients was carried out using a nomogram and calibration plots, which indicated that CASP4 could predict the overall survival of GBM patients. Therefore, this nomogram could be used to provide a more accurate survival prognostic judgment on GBM patients. These results demonstrated that CASP4 was a potential prognosis factor for GBM. The current GBM prognosis is abysmal, with a median survival time of 12–15 months under standard treatments [24]. Recent evidence from clinical trials showed that targeted therapy does not improve the prognosis of patients with GBM [25]. TMB has been reported to influence immunotherapeutic effectiveness across tumor types and can be used to predict the survival of diverse tumors with CTLA-4 or anti-PD-1 treatments [26]. Therefore, these PRGs are highly expressed in GBM and their high expression is related to poor survival and disease progression. They also correlate with tumor infiltration by immune cells and immune therapy indicators such as TMB genes. PRGs offer valuable GBM biomarkers for prognostic and immune therapy response evaluations. 

The induction of other PCDs, such as necroptosis and pyroptosis, may improve chemotherapy performance as other PCDs could overcome apoptosis [18]. Upregulated genes are involved in protein folding (e.g., HSP90AA1 and HSP90AB1) and molecular chaperones (e.g., HSPB1 and CRYAB) are also associated with autophagy, apoptosis, and generalized stress responses [27]. Autophagy is an additional form of cell death to those previously mentioned. Autophagy plays a complex role in tumor development; it is triggered by ROS and is cross-linked with apoptosis and pyroptosis to promote or inhibit apoptosis under different microenvironmental conditions [28]. Therefore, natural products with a dual role of autophagy inhibition and pyroptosis induction could be an ideal adjuvant to chemotherapy. Previous research showed that kaempferol could decrease the mitochondrial membrane potential and increase ROS in glioma cells, which can induce autophagy and subsequently trigger pyroptosis, indicating that kaempferol has a clinical potential as a natural molecule against GBM [28]. Baeomycesic acid, derived from the lichen *Thamnolia*, is reported to have an anti-inflammatory activity and is specifically active against 5-LOX [29]. As the main compound in feverfew, parthenolide has been used to cure migraines and rheumatoid arthritis for a long time. It induces apoptosis in human cancer cells, with its gene enrichment showing that it mainly regulates apoptosis [30]. So, parthenolide can be used as a cooperating pharmaceutical agent for the cancer chemotherapy of various malignancies. Combining drugs with multiple targets and MOAs can inhibit cancer cell proliferation or reduce metastasis under cooperation with parthenolide [31]. In this research, parthenolide could suppress the expression of most subtype cells, including endothelial cells, microglia, macrophages, T cells, dendritic cells, etc. Therefore, parthenolide could function as a lead molecule in a combined therapy for GBM. Rutin (quercetin glycoside) is a natural product that is widely found in fruit and vegetables and has significant anticancer effects [32]. However, its anticancer mechanisms have not been clearly elucidated. Nevertheless, its enrichment analysis showed responses to oxidative stress. Luteolin also has an anti-inflammatory and neuroprotective effect *via* the inhibition of NF-kB and MAPK activation [33]. 

Our study aimed to support the use of natural lead molecules as adjuvants with chemotherapeutic drugs for GBM treatment. Precision/personalized herbal medicines are both timely and essential for modern therapeutics due to unsatisfactory clinical outcomes, defective treatment strategies, and adverse effects to existing drugs. In addition, biomarker innovations stand the test of real-life practice and their implementation in clinical settings and societies. Therefore, the strategy developed in this study could serve as a model for personalized chemotherapy with natural products as treatment adjuvants that have an excellent therapeutic effect and low toxic side effects.

## 4. Materials and Methods 

A workflow chart based on an integrative strategy of pharmacogenomic analyses to investigate combined therapies is shown in Figure 7.

### 4.1. Single-Cell Sequencing Analysis

The scRNA-seq raw sequencing data of GSE162631 were downloaded from the GEO database [34]. These included eight sample datasets, including (R1–4) tumor cores (GSM4955731, GSM4955733, GSM4955735, and GSM4955737) and tumor peripheral tissue (GSM4955732, GSM4955734, GSM4955736, and GSM4955738). These samples were analyzed using GPL24676 Illumina NovaSeq6000 (Homo sapiens). These datasets for the eight samples were analyzed using the R package (Seurat) [35]. The percentage of mitochondrial genes was calculated to collect high-quality single-cell data with a high fraction of reads in cells (80%) and a low fraction of cells enriched in mitochondrial genes (5%). The cells were filtered after analyzing the distribution correlation between the genes, mitochondrial genes, and RNAs. A standardized gene expression was used to calculate the differentially expressed genes using the R function (FindVariableFeatures) [36]. The normalized data were used to perform principal component analysis (PCA) and subsequently plotted by a shared nearest-neighbor algorithm (t-SNE), which colored various clusters with specific features to visualize the gene expression. The differentially expressed genes (DEGs) between subpopulations were presented in annotated matrices and the top 16 DEGs related to each cluster were used to plot a heatmap. The Seurat function (FindAllMarker) was used to identify a set of signature marker genes for each cluster compared with all other cells [37]. The top four cell markers were based on each cluster’s expression level (15 clusters in the t-SNE figure) to show these four genes with cell-specific expressions (Appendix A). Each cluster’s cell type was identified by searching the CellMarker database [38]. 

### 4.2. Pyroptosis Gene Analysis

The pyroptosis-related genes were collected from The Human Gene Database (GeneCards) with screening criteria of a relevance score > 1 (102 genes). GeneCards is a comprehensive database of functions involving proteomics, genomics, and transcriptomics [39]. The intersection of DEGs associated with the GBM single-cell sequencing dataset and pyroptosis-related genes (PRGs; 102) was determined as GBM-related pyroptosis genes. These intersected genes could be used in subsequent prognostic prediction, immune abundance, and drug prediction analyses. The mutation frequencies and oncoplot waterfall plots of the 30 intersected PRGs in the GBM scRNA-seq samples were generated by analyzing the expression distribution of the mRNA of 30 PRGs in tumor and normal tissues. For the GBM patients from the TCGA database, tumoral RNA-seq data were downloaded from the Genomics Data Commons (GDC) data portal (TCGA) and the paired normal tissue samples were downloaded from the same portal. Somatic datasets and copy number variation (CNV) data for GBM were also downloaded from TCGA and the University of California Santa Cruz (UCSC) Xena website. The genetic mutation data and clinical data from the TCGA database were downloaded to identify the somatic mutations of the PRGs. Mutation data were downloaded and visualized using the matfools package in R (R Foundation for Statistical Computing, Vienna, Austria, RRID:SCR_003302). A horizontal histogram showed that the genes had a higher mutation frequency in GBM patients [40,41]. The PRG-TF-miRNA regulatory network was constructed using the miRNet database, an online tool with information generated from miRNA investigations. This database is associated with various miRNA databases such as TarBase and miRTarBase. Briefly, 30 PRGs were uploaded to the query box of GENES, specifying the organism (Homo sapiens), ID type (official gene symbol), tissue (human-only, with brain), which were targeted by miRNA and TF in the database of TRUST. The protein–protein interaction (PPI) network of PRGs was built using the STRING database.

### 4.3. The Prognostic Value of Pyroptosis-Related Genes

Raw counts of RNA-seq data and corresponding clinical information were downloaded from the TCGA database. The survival difference of PRGs between the above two groups was compared using a log-rank test with a Kaplan–Meier (KM) survival analysis. For the KM curves of the 30 PRGs, *p*-values and hazard ratios (HRs) with a 95% confidence interval (CI) were created by a univariate Cox proportional hazard regression using the R packages survival and survminer [42]. The top eight genes with the highest hazard ratio values were selected to build the prognostic model based on the least absolute shrinkage and selection operator (LASSO) regression algorithm for the feature selection using a ten-fold cross-validation via the R package glmnet [43]. Nomogram, another predictive prognostic model based on biomarkers and clinical parameters (i.e., age, gender, and stage), was used to perform the univariate and multivariate Cox regressions. The forest plots showed the *p*-value, HR, and 95% CI of each variable obtained using the R package forestplot [44]. The nomogram was developed based on multivariate Cox proportional hazard analysis results to predict 1-, 2-, and 3-year overall recurrence. The factors used to calculate the recurrence risk for individual patients were visualized as a graphical presentation *via* the R package rms [45]. The closer the nomogram model was to the calibration curve, the better the model prediction results.

### 4.4. The abundance of Immune Cells and Tumor Mutation Burden in GBM

The association between prognostic PRGs and immune infiltration was investigated using the Tumor Immune Estimation Resource (TIMER) based on the LASSO model with a risk score. According to the model, the higher the score, the greater the risk. Several immune cell markers in TIMER and Gene Expression Profiling Interactive Analysis (GEPIA) were used to characterize immune cells, including B cells, CD8^+^ T cells, CD4^+^ T cells, macrophages, neutrophils, and myeloid dendritic cells [28]. In addition, the correlation between PRGs and tumor mutation burden (TMB) was examined using Spearman’s calculation between gene expression and TMB using the R package ggstatsplot. A *p*-value of less than 0.05 was considered to be statistically significant [46,47]. 

### 4.5. Drug Prediction, Drug Sensitivity, and Drug Validation

The drug prediction for GBM treatment was carried out using Enrichr platforms based on the GBM-related PRGs (30 genes); an adjusted *p*-value threshold of 0.05 was used to screen the repurposed drugs. A Sankey diagram was constructed based on the results of Enrichr with the *p*-value, gene ratio, and gene counts calculated via the R package ggaluval [48]. The top 300 DEGs, which included 150 upregulated and 150 downregulated genes ranked by the absolute log 2 or fold change values in descending order, were selected to query CMap for therapeutic agents. The connectivity scores in the result of the CMap query ranged from −1 to 1, indicating the similarity of the gene expression profile between the query signature and the CMap instance. Among the scores, a positive connectivity score indicated the similar induction of an expression change by a query signature from diseased samples compared with the pertubagen in the CMap database, which could serve as an inducer. In contrast, the therapeutic agents could be selected from those with high negative connectivity scores because they displayed reverse effects on the gene expression [49]. The top-scoring 21 natural products with the highest relevance scores from the CMap results were selected to query their targets via the SymMap platform or to predict their targets using the PharmMapper Server or Similarity ensemble approach (SEA) Server, based on their 3D structures downloaded from PubChem. The targets of these 21 natural products were intersected with each cell type’s top 50 signature genes (15 clusters from the scRNA-seq GBM dataset). The intersected results were used to build a Sankey diagram to show these 21 natural products’ combined therapy on 15 different cell types of GBM. 

## 5. Conclusions

Our study identified five top-scoring natural products: parthenolide, rutin, baeomycesic acid, luteolin, and kaempferol. These natural products exhibited different mechanisms of action, including NF-kB inhibition, antioxidant activity, lipoxygenase inhibition, glucosidase inhibition, and estrogen receptor agonism, respectively. Furthermore, our analysis of cell-type-specific differential expression-related targets revealed that the top five subtype cells targeted by natural compounds in GBM were endothelial cells, microglia/macrophages, T cells, dendritic cells, and neutrophils. Notably, parthenolide demonstrated the ability to suppress the expression of most subtype cells, including endothelial cells, microglia, macrophages, and dendritic cells. Therefore, parthenolide shows promise as a lead molecule in a combined therapy for GBM cases. Our findings support the potential of a polypharmacological approach using natural lead molecules to overcome chemotherapy drug resistance and enhance the effectiveness of GBM treatment. This approach offers a new model for the development of personalized therapeutic strategies and identifying first-in-class clinical leads. It can also contribute to the emergence of biomarker-informed personalized herbal medicine. However, it is important to acknowledge the limitations of our study; further experiments are needed to validate the efficacy and safety of these natural lead molecules in combination with chemotherapeutic drugs. Nonetheless, our study provides valuable insights and lays the foundation for future innovative research on combined therapies against GBM.

## Figures and Tables

**Figure 1 pharmaceuticals-16-01533-f001:**
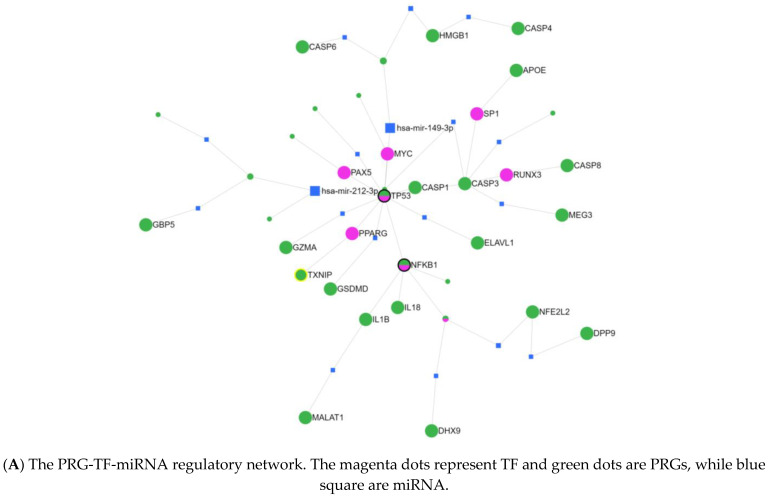
Landscape of genetic and expression variations of pyroptosis-related genes in GBM. (**A**) The PRG-TF-miRNA regulatory network of 30 PRGs using the miRNet database. (**B**) The gene expression profile across all tumor samples and paired normal tissues (dot plot). (**C**) The lollipop plot of the mutation distribution and protein domains of TP53 in cancer. (**D**,**E**) The mutation frequency and classification of TP53 from the TCGA GBM cohort. The middle panel shows the variant classification summary in the cohort, using the same mutation-specific color code. (**F**) The expression of 30 PRGs in three different brain tissues. The horizontal axis represents PRGs. The vertical axis represents the PRG expression distribution, where different colors represent different groups (normal, GBM, and LGG). The upper left corner represents the significant *p*-value of the Wilcox test method.

**Figure 2 pharmaceuticals-16-01533-f002:**
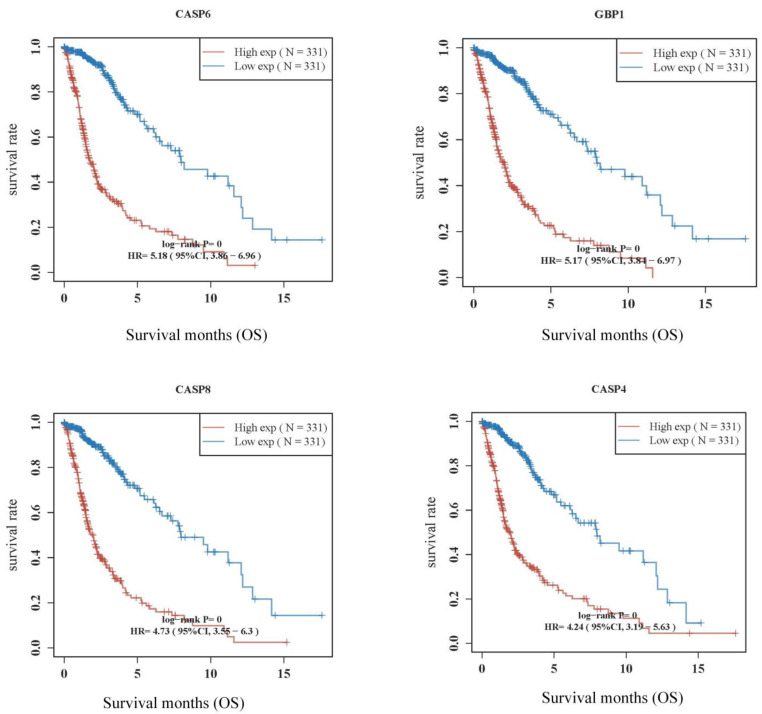
The prognostic value of pyroptosis-related genes with top 8 HR values of GBM patients in the high-/low-expression groups. A log-rank test was used to compare differences in survival between these groups. For Kaplan–Meier curves, log-rank tests and univariate Cox proportional hazard regressions generated *p*-values and hazard ratios (HRs) with a 95% confidence interval (CI). The Kaplan–Meier curve shows the cumulative survival probabilities. A steeper slope indicates a higher event rate (death rate) and a worse survival prognosis. A flatter slope indicates a lower event rate and a better survival prognosis.

**Figure 3 pharmaceuticals-16-01533-f003:**
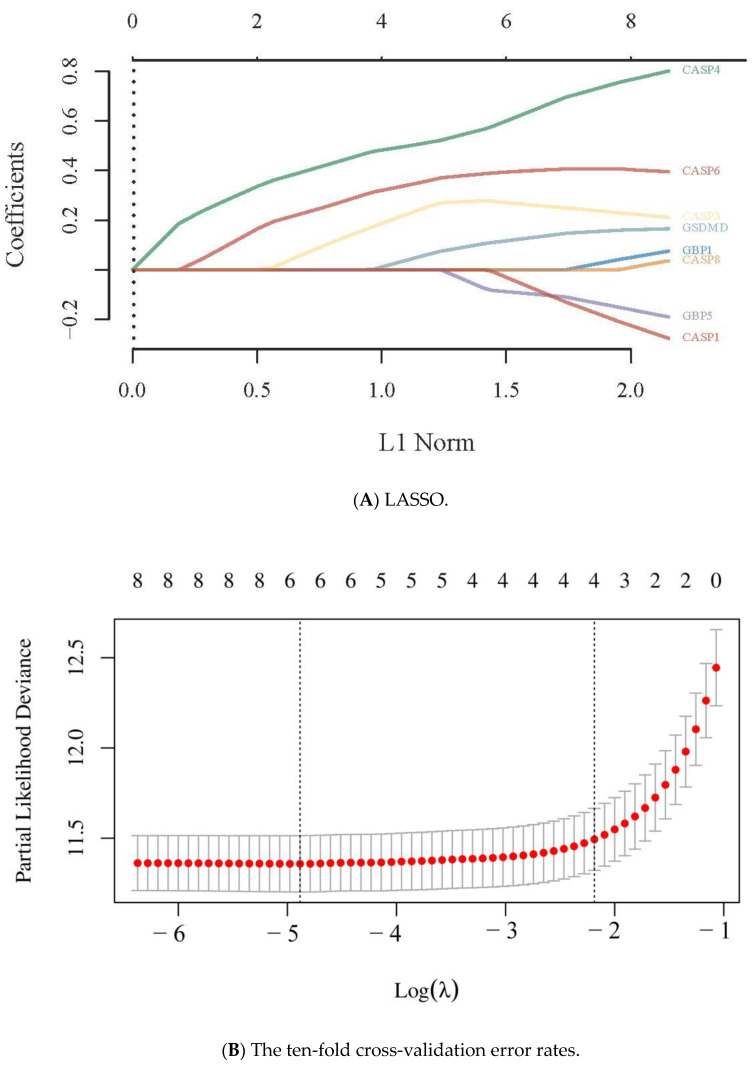
Construction of a prognostic pyroptosis-related gene model. (**A**) LASSO coefficient profiles of the eight pyroptosis-related genes. (**B**) Plots of the ten-fold cross-validation error rates. (**C**) Distribution of risk score, survival status, and expression of top seven prognostic pyroptosis genes in GBM. (**D**,**E**) Overall survival curves for GBM patients in the high-/low-risk group and ROC curve used to measure the predictive value.

**Figure 4 pharmaceuticals-16-01533-f004:**
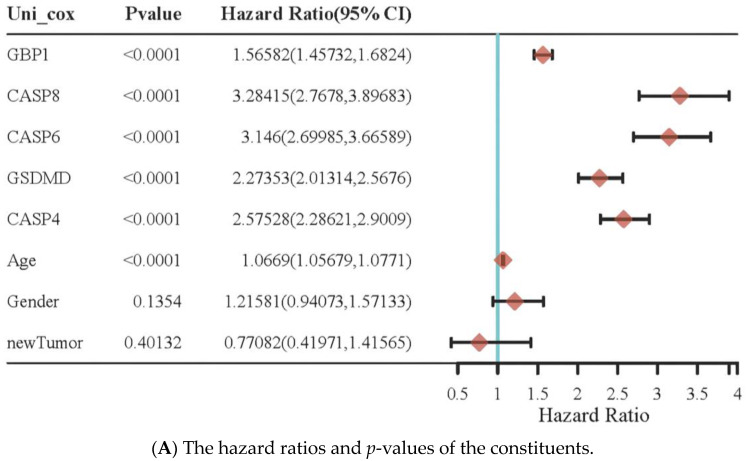
Construction of a predictive nomogram. (**A**,**B**) Hazard ratios and *p*-values of the constituents involved in univariate and multivariate Cox regressions considering clinical parameters and top 5 pyroptosis-related genes in GBM. (**C**,**D**) Nomogram to predict the 1-year, 2-year, and 3-year overall survival rate of GBM patients. Calibration curve for the overall survival nomogram model in the discovery group. A dashed diagonal line represents the ideal nomogram.

**Figure 5 pharmaceuticals-16-01533-f005:**
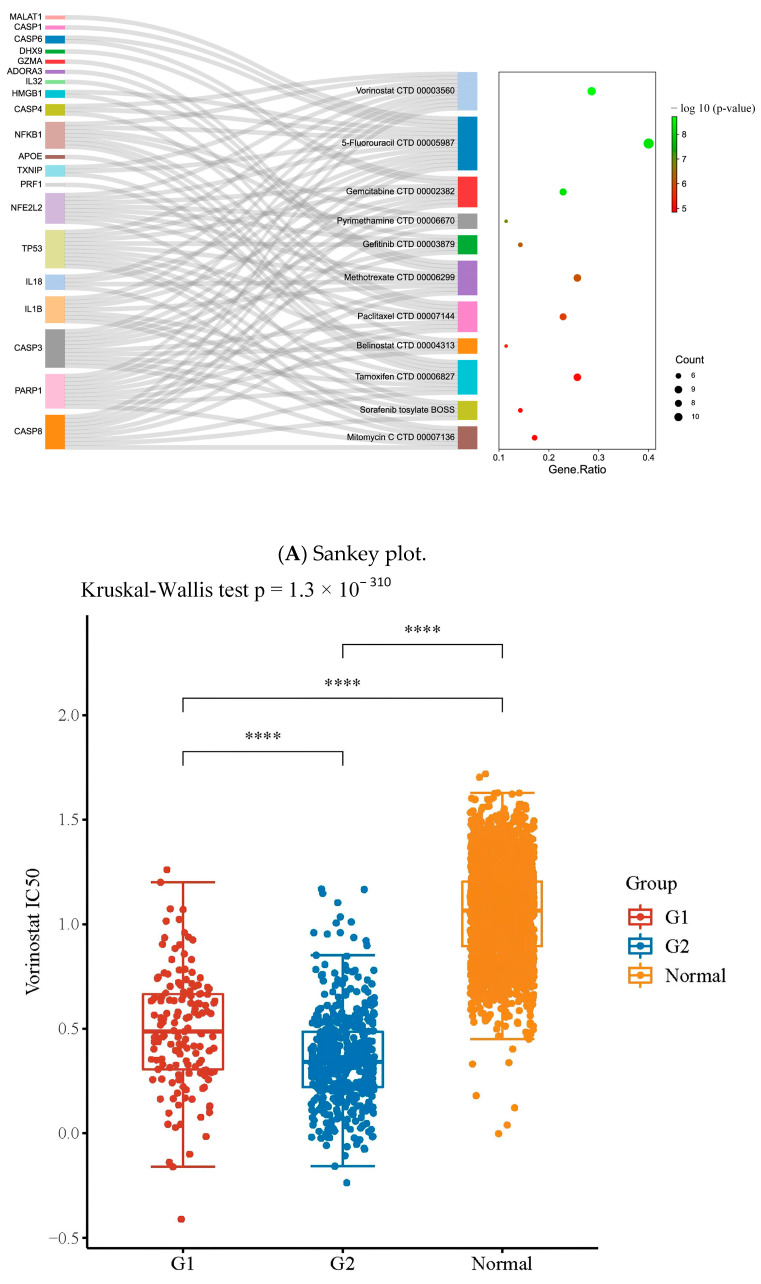
Cancer therapeutic drug prediction from the relationship between diseases and the drug database (DSigDB) as well as IC50 analyses of these repurposed drugs. (**A**) Sankey plot showcasing that the drugs repurposed to the pyroptosis-related phenotype of GBM (20 out of 30 genes) were involved in each repurposed drug obtained via the DSigDB database. In addition, the dot plot shows the ratio between pyroptosis-related genes specific to repurposed drugs and the total number of genes targeted by each drug (FDR; *p* < 0.05). (**B**) The GDSC drug sensitivity analysis of GBM (IC50). ns means *p* > 0.05, *** represents *p* ≤ 0.001 and **** *p* ≤ 0.0001.

**Figure 6 pharmaceuticals-16-01533-f006:**
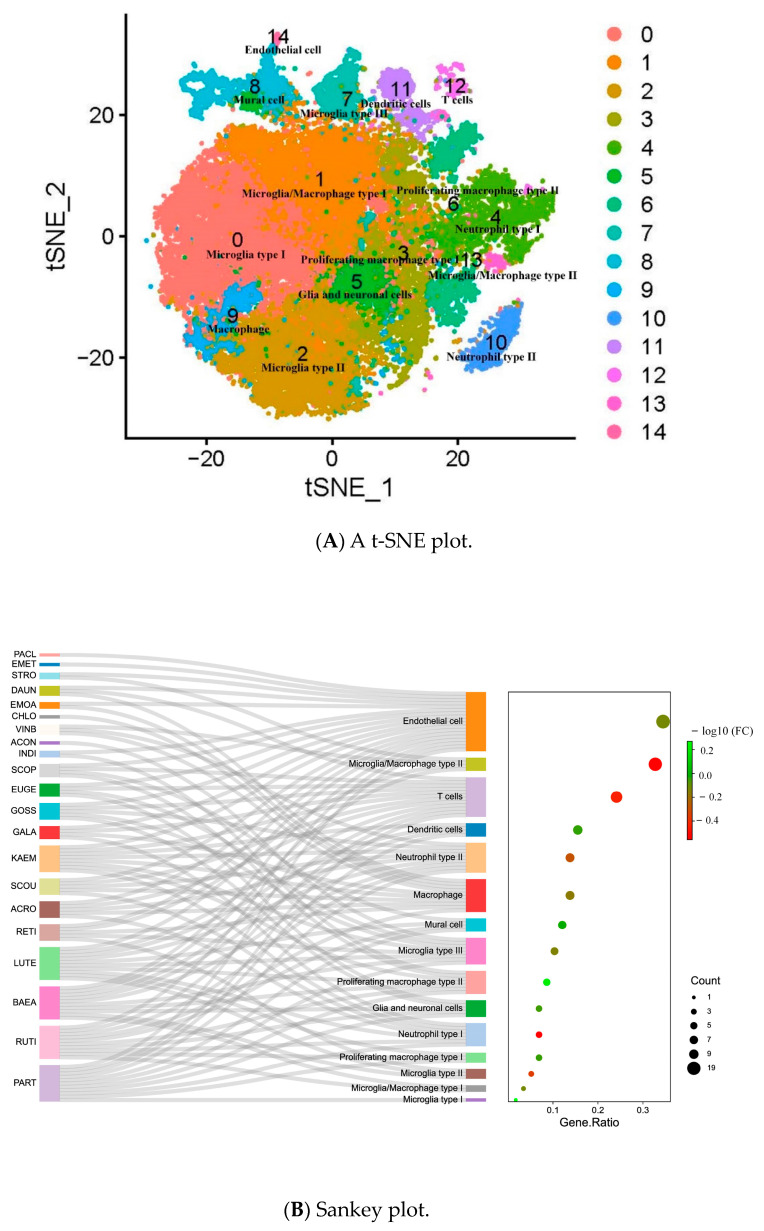
The association between natural products from CMap with their targeted subtype cells. (**A**) A t-SNE plot visualizing single-cell RNA-seq data of 8 GBM samples with 97,584 cells. (**B**) Sankey plot showcasing natural products specific to the subtype cells of GBM samples. The dot plot shows the gene ratio of each subtype cell targeted by natural products (*p* < 0.05). PART: parthenolide; RUTI: rutin; BAEA: baeomycesic acid; LUTE: luteolin; RETI: retinol; ACRO: acronycine; SCOU: scopolamine; KEAM: kaempferol; GALA: galangin; GOSS: gossypol; EUGE: eugenol; SCOP: scopolamine; INDI: indirubin; ACON: acronycine; VINB: vinblastine; CHLO: chloroquine; EMOA: emodic acid; DAUN: daunorubicin; STRO: strophanthidin; EMET: emetine; PACL: paclitaxel.

**Figure 7 pharmaceuticals-16-01533-f007:**
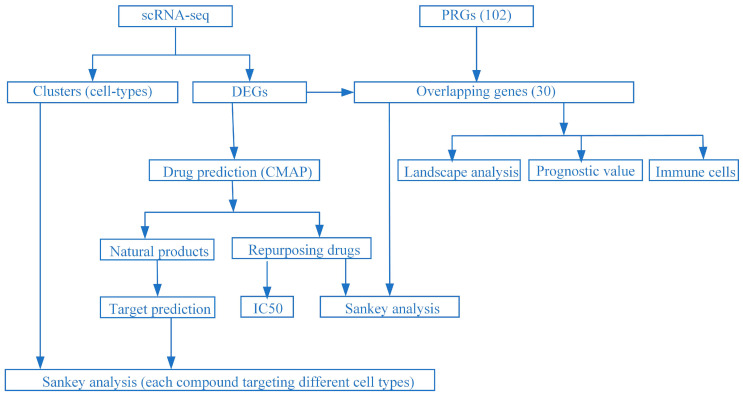
Framework based on an integrative strategy of pharmacogenomic analyses to investigate combined therapies.

**Table 1 pharmaceuticals-16-01533-t001:** The top-scoring five natural compounds selected from CMap.

No.	Name of Natural Products	Structures
1	Parthenolide (PubChem CID 7251185) (ADMET_BBB_Level: good)	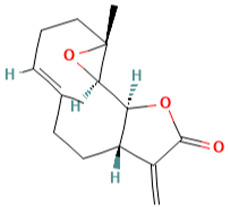
2	Rutin (PubChem CID 5280805) (ADMET_BBB_Level: low)	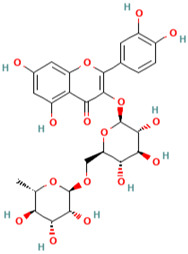
3	Baeomycesic acid (PubChem CID 5321461) (ADMET_BBB_Level: low)	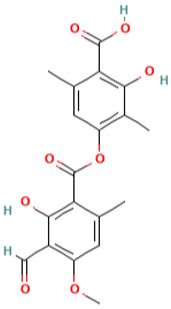
4	Luteolin (PubChem CID 5280445) (ADMET_BBB_Level: low)	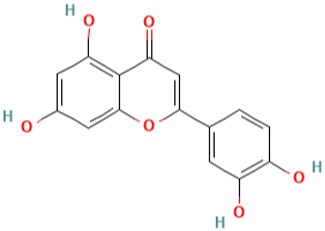
5	Kaempferol (PubChem CID 5280863) (ADMET_BBB_Level: low)	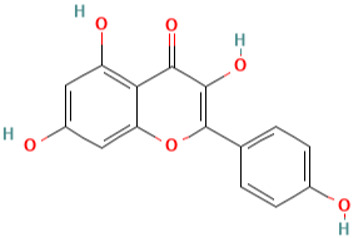

## Data Availability

Data is contained within the article and Appendix A.

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
