# Peer review of "Pharmacogenomic Analysis of Combined Therapies against Glioblastoma Based on Cell Markers from Single-Cell Sequencing"

_pharmaceuticals, 2023, doi:10.3390/ph16111533_

Round 1

Reviewer 1 Report

Liu and colleague’s research intends to identify biomarkers to monitor the course of illness and to develop personalised preventative and therapeutic solutions for glioblastoma. The hub gene in the pyroptosis process in glioblastoma (GBM) was TP53, as anticipated in the pyroptosis-related genes-transcription factor-microRNA regulation network. The one-, two-, and three-year overall survival rates of GBM patients were correctly and simply predicted using the LASSO-Cox regression model of pyroptosis-related genes. Parthenolide, rutin, baeomycesic acid, luteolin, and kaempferol, which have antioxidant, NFKB inhibition, lipoxygenase inhibition, glucosidase inhibition, and oestrogen receptor agonism, respectively, are the top five natural chemicals. Contrarily, the examination of the targets of natural substances connected to cell type-specific differential expression revealed that the top 5 subtypes.

The work is generally well written, the techniques are well explained, and the results are thoroughly discussed. Although the study given is not ground breaking, the models offered in the paper are a significant contribution to the knowledge base. The data presented in the paper supports the conclusions. 

Before accepting this article, I have few minor concerns to be addressed

The authors should present a flow diagram of overall methodology/analysis employed in the manuscript that can help the readers to grasp the overall direction of the paper.

The quality of the figure should be improved and the symbols in the figure should be enlarged

The conclusion should be written more clearly and strictly summarize the results.

The authors miss some important citations (PMIDs: 33140689, 20616955)

Author Response

Liu and colleague’s research intends to identify biomarkers to monitor the course of illness and to develop personalised preventative and therapeutic solutions for glioblastoma. The hub gene in the pyroptosis process in glioblastoma (GBM) was TP53, as anticipated in the pyroptosis-related genes-transcription factor-microRNA regulation network. The one-, two-, and three-year overall survival rates of GBM patients were correctly and simply predicted using the LASSO-Cox regression model of pyroptosis-related genes. Parthenolide, rutin, baeomycesic acid, luteolin, and kaempferol, which have antioxidant, NFKB inhibition, lipoxygenase inhibition, glucosidase inhibition, and oestrogen receptor agonism, respectively, are the top five natural chemicals. Contrarily, the examination of the targets of natural substances connected to cell type-specific differential expression revealed that the top 5 subtypes.

The work is generally well written, the techniques are well explained, and the results are thoroughly discussed. Although the study given is not ground breaking, the models offered in the paper are a significant contribution to the knowledge base. The data presented in the paper supports the conclusions.

Response: Thank you very much for your positive comments on our work.

Before accepting this article, I have few minor concerns to be addressed

The authors should present a flow diagram of overall methodology/analysis employed in the manuscript that can help the readers to grasp the overall direction of the paper.

Response: great suggestion. We have created a workflow chart about the analysis procedure (see Figure 1).

The quality of the figure should be improved and the symbols in the figure should be enlarged.

Response: We improved the figure quality.

The conclusion should be written more clearly and strictly summarize the results.

Response: We rewrote the conclusion as you suggested.

The authors miss some important citations (PMIDs: 33140689, 20616955)

Response: We added these two references to our manuscript as follows.

  1. Mutharasu G, Murugesan A, Konda Mani S, Yli-Harja O, Kandhavelu M. Transcriptomic analysis of glioblastoma multiforme providing new insights into GPR17 signaling communication. J Biomol Struct Dyn. 2022 Apr;40(6):2586-2599. doi: 10.1080/07391102.2020.1841029. Epub 2020 Nov 3. PMID: 33140689.
  2. Moustakas A, Kreisl TN. New treatment options in the management of glioblastoma multiforme: a focus on bevacizumab. Onco Targets Ther. 2010 Jun 24;3:27-38. doi: 10.2147/ott.s5307. PMID: 20616955; PMCID: PMC2895775.

Reviewer 2 Report

The manuscript is in general good, but needs to summarized and concentrated.

good

Author Response

The manuscript is in general good, but needs to summarized and concentrated.

Response: thank you very much for your positive comments. We concentrated it as you suggested.

Reviewer 3 Report

Comments and Suggestions for Authors of Pharmaceuticals-2605695_r1:

The authors of this manuscript predicted several pyroptosis-associated drugs for GBM through the combination of scRNA-seq analysis of GBM, the LASSO-Cox regression model, and several bioinformatics tools and databases including CMap and GDSC.

Suggestions:

·        In Materials and Methods 2.1 section, the authors should provide some content, datasets, and methods for this scRNA-seq analysis. If it were just a quote, you would not even list it in this section, and then you should provide clearly the resource. 

Our readers would understand better, if the authors added a workflow of the entire study in this section, it would be better for readers to understand this manuscript.

·         In Results section, the authors should explain the connection with other parts of the manuscript and provide the method to obtain Fig. 1 A. In addition, the authors should provide and list the original 185 PRG and 130 selected PRG at least in the supplements.

·         In Fig. 2, the authors should explain the label of x-axis, ‘survival years’, because the median survival time of GBM patients is about 15 months after diagnosis.  Please check and explain all about survival time.

·         In the figure legend of Fig. 5, the authors should not explain anything about IC50 or GDSC; these should be shown in the content, and they are already there.

·         There are insufficient validation studies for the drugs predicted by CMap, and it is recommended to use CMap2 to assess their comparability and reliability.

The quality of English can be proved.

Author Response

Comments and Suggestions for Authors of Pharmaceuticals-2605695_r1:

The authors of this manuscript predicted several pyroptosis-associated drugs for GBM through the combination of scRNA-seq analysis of GBM, the LASSO-Cox regression model, and several bioinformatics tools and databases including CMap and GDSC.

Suggestions:

 In Materials and Methods 2.1 section, the authors should provide some content, datasets, and methods for this scRNA-seq analysis. If it were just a quote, you would not even list it in this section, and then you should provide clearly the resource.

Response: thank you for the suggestion. We have clarified the contents such as the dataset name, and methods more in the text.

Our readers would understand better, if the authors added a workflow of the entire study in this section, it would be better for readers to understand this manuscript.

Response: We created a workflow chart to clearly show the analysis procedure. Thank you.

In Results section, the authors should explain the connection with other parts of the manuscript and provide the method to obtain Fig. 1 A. In addition, the authors should provide and list the original 185 PRG and 130 selected PRG at least in the supplements.

Response: We put more details in the methods about how to create Fig.1A. We tried to find out the important microRNA regulatory network since MiRNA is expected to regulate more than 60% of protein-coding genes and participate in nearly all known cellular processes, such as cell cycle, differentiation, cell proliferation, apoptosis, etc. In recent years, it has become an indisputable fact that miRNA is involved in the occurrence of tumors. A large number of studies have also confirmed that miRNA has dual functions of suppressing and promoting tumors, we hoped to come up with a new idea for tumor research to construct this regulatory network. We have listed the original PRG in the Table 1 in the supplement materials.

In Fig. 2, the authors should explain the label of x-axis, ‘survival years’, because the median survival time of GBM patients is about 15 months after diagnosis.  Please check and explain all about survival time.

Response: thank you for the correction. We have corrected and explained the survival time and calculation.

In the figure legend of Fig. 5, the authors should not explain anything about IC50 or GDSC; these should be shown in the content, and they are already there.

Response: we deleted the description of IC50 and GDSC.

There are insufficient validation studies for the drugs predicted by CMap, and it is recommended to use CMap2 to assess their comparability and reliability.

Response: we tried a couple of times to register CMAP 2 but failed. Maybe we will try to do it next time.

Reviewer 4 Report

This is an interesting study, and the authors have provided a detailed analysis of their work. The aim of the study is quite clear in the abstract and materials and methods are well explained in detail, making it easy to reproduce the results. Results are referred to statistical analysis methods and referenced accordingly. All ethical standards for data collection are met. Finally, the conclusion is short and crisp, summarizing the research and confirming the aim and need for this research. There were a few minor grammatical and graph labeling errors. Some of the suggested revisions are as follows: 

1. Abstract: Please elaborate a little on “prophylactic and therapeutic strategies, showing their relevance in the study.

L 21: Kindly rephrase “pyroptosis-related genes-transcription factor-microRNA regulatory network” as it is lengthy and challenging to follow.

2. Introduction: L 42-43: Ending the sentence “represents a huge unmet clinical need” is abrupt and unclear. Kindly specify precisely how and what the clinical need is that is not being met.

3. L 55: Kindly omit “fully” at the end of the sentence. This sentence is again repeated in L 58. Please use it once, probably in L 58, and omit the sentence from L 53.

4. L 71: Please add proper punctuation marks - “anticancer drugs, target proteins, or essential functions”. Overall, this sentence is unclear, and it seems some information has been missed. The sentence aims to inform about challenges of developing specific inhibitors to counteract the detrimental impacts of mutated oncogenic proteins but fails to clearly tell us what the challenges are.

5. Materials and Methods:

Pyroptosis-related genes should be mentioned with its acronym (PRGs) at the start of the paragraph. Its sudden usage from L 127 onward is not correct.

6. L 139: Kindly change “such TarBase” to “such as TarBase”.

7. Results:

L 231: Kindly change “showed the high expression” to “showed high expression”.

8. P6 Fig-1 B): Kindly label the normal tissues and tumor samples in the DotPlot. It looks incomplete. Similarly, Fig-1 E) has no labels or clear titles. Please add them for better clarity and understanding.

9. L 264-265: The sentence should be rephrased to avoid overlapping of information and provide better understanding. It could be changed to - “of the two groups corresponding to 50% of the time in units of years. For example, the overall survival times in the high and low expression groups are 1.7 years and 8 years, respectively.”

10. P11 Fig-3 C): In the graph showing the expression of 8 prognostic pyroptosis genes in GBM, there are only 7 genes mentioned. I believe CASP8 is missing. Kindly check the graph and revise it accordingly. Moreover, Fig-3 E) is not labeled, though it does have a description. Please label it.

11. L 345: Kindly add “show that” after GBM in the sentence.

L 427: Kindly change “is inconsistent” to “is consistent”.

L 457: Kindly change “Thamnolia is reported” to “Thamnolia, reported”.

L 476: Kindly omit the use of “that” after biomarker innovations.

12. Cite the following reference in the introduction section: https://doi.org/10.3390/medicina59030514

Author Response

This is an interesting study, and the authors have provided a detailed analysis of their work. The aim of the study is quite clear in the abstract and materials and methods are well explained in detail, making it easy to reproduce the results. Results are referred to statistical analysis methods and referenced accordingly. All ethical standards for data collection are met. Finally, the conclusion is short and crisp, summarizing the research and confirming the aim and need for this research. There were a few minor grammatical and graph labeling errors. Some of the suggested revisions are as follows:

Response: we have modified the conclusion to make it much more clear.

  1. Abstract: Please elaborate a little on “prophylactic and therapeutic strategies, showing their relevance in the study.

Response: we tried to find a new and stable biomarker which can be used as a sensitive diagnosis of GBM.

L 21: Kindly rephrase “pyroptosis-related genes-transcription factor-microRNA regulatory network” as it is lengthy and challenging to follow.

Response: PRGs (pyroptosis-related genes)-TF (transcription factor)-MiRNA (microRNA) regulatory network.

  1. Introduction: L 42-43: Ending the sentence “represents a huge unmet clinical need” is abrupt and unclear. Kindly specify precisely how and what the clinical need is that is not being met.

Response: We modified the sentence like this: A huge unmet clinical need is the full understanding of the pathogenesis and related pathways of GBM.

  1. L 55: Kindly omit “fully” at the end of the sentence. This sentence is again repeated in L 58. Please use it once, probably in L 58, and omit the sentence from L 53.

Response: Thank you for the correction. We have deleted the whole repeated part.

  1. L 71: Please add proper punctuation marks - “anticancer drugs, target proteins, or essential functions”. Overall, this sentence is unclear, and it seems some information has been missed. The sentence aims to inform about challenges of developing specific inhibitors to counteract the detrimental impacts of mutated oncogenic proteins but fails to clearly tell us what the challenges are.

Response: The big challenge is the drug targets expressed in both normal and cancer cells.

  1. Materials and Methods:

Pyroptosis-related genes should be mentioned with its acronym (PRGs) at the start of the paragraph. Its sudden usage from L 127 onward is not correct.

Response: Thank you for the correction. We modified it.

  1. L 139: Kindly change “such TarBase” to “such as TarBase”.

Response: We modified it.

  1. Results:

L 231: Kindly change “showed the high expression” to “showed high expression”.

Response: we deleted it.

  1. P6 Fig-1 B): Kindly label the normal tissues and tumor samples in the DotPlot. It looks incomplete. Similarly, Fig-1 E) has no labels or clear titles. Please add them for better clarity and understanding.

Response: We modified it as you suggested to make it clear.

  1. L 264-265: The sentence should be rephrased to avoid overlapping of information and provide better understanding. It could be changed to - “of the two groups corresponding to 50% of the time in units of years. For example, the overall survival times in the high and low expression groups are 1.7 years and 8 years, respectively.”

Response: Thank you for the corrections.

  1. P11 Fig-3 C): In the graph showing the expression of 8 prognostic pyroptosis genes in GBM, there are only 7 genes mentioned. I believe CASP8 is missing. Kindly check the graph and revise it accordingly. Moreover, Fig-3 E) is not labeled, though it does have a description. Please label it.

Response: we checked the graph with the data and R package, and it seems like the CASP8 with negligible effect (0.0004) on the lasso and also see the coefficient in Fig 4A, so the graph and lasso equation did not include it.

  1. L 345: Kindly add “show that” after GBM in the sentence.

Response: we added it to the sentence.

L 427: Kindly change “is inconsistent” to “is consistent”.

Response: thank you for the correction.

L 457: Kindly change “Thamnolia is reported” to “Thamnolia, reported”.

Response: We corrected.

L 476: Kindly omit the use of “that” after biomarker innovations.

Response: we corrected it.

  1. Cite the following reference in the introduction section: https://doi.org/10.3390/medicina59030514

Response: we added this reference to the introduction.

Round 2

Reviewer 3 Report

I am satisfied this version. 

Reviewer 4 Report

The said manuscript has been revised as per my comments & suggestions.